# Examining the Economic Value of Tourism and Visitor Preferences: A Portrait of Sustainability Ecotourism in the Tangkahan Protection Area, Gunung Leuser National Park, North Sumatra, Indonesia

**Agus Purwoko [1,*], Dodik Ridho Nurrochmat [2], Meti Ekayani [3], Syamsu Rijal [4] and Herlina Leontin Garura [1]**

1   Faculty of Forestry, Universitas Sumatera Utara, Jl. Tri Dharma Ujung No.1 Kampus USU Medan, Medan City 20155, North Sumatra, Indonesia; herlinaleontinn@gmail.com
2   Faculty of Forestry and Environment, IPB University, Jl. Lingkar Akademik IPB Dramaga Campus, Bogor 16680, West Java, Indonesia; dnurrochmat@ipb.ac.id
3   Faculty of Economics and Management, IPB University, Jl. Agatis, IPB Dramaga Campus, Bogor 16680, West Java, Indonesia; meti@ipb.ac.id
4   Faculty of Forestry, Universitas Hasanuddin, Jl. Perintis Kemerdekaan No. Km.10, Tamalanrea Indah, Makassar 90245, South Sulawesi, Indonesia; jsyamsurijal@unhas.ac.id
*   Correspondence: agus9@usu.ac.id; Tel.: +62-813-6051-3501

**Abstract:** North Sumatra Province has the Tangkahan Nature Tourism Area, which represents ecotourism managed by local communities, established in 2001, which has now become the leading tourism destination of North Sumatra both locally and internationally. Tangkahan ecotourism is an example of payment for environmental services for the Tangkahan community, which initially carried out illegal logging in the mount Leuseur national park and then agreed to preserve the national park through ecotourism. This study aims to analyze the economic value of tourism and the preferences of tourists to revisit, along with the factors that influence them, where these conditions can be an illustration of the sustainability of Tangkahan ecotourism. The travel cost method is used to calculate the economic value of Tangkahan Ecotourism environmental services. The factors that affect the economic value, intensity of visits, and interest in revisiting, were analyzed using multiple linear regression. The results showed that Tangkahan ecotourism has a relatively high economic value, supported by the intensity and interest of tourist visits. Factors that affect the economic value and preferences of tourist visits can be managed for the sustainability of Tangkahan ecotourism so as not to lose the economic value of the ecotourism environmental services.

**Keywords:** ecotourism; Tangkahan; economic value; intensity of visits; travel cost method; interest in revisiting

## 1. Introduction

Forest resources have various interests that should be considered optimally. These interests are fragmented into interests of the community, law enforcement, conservation goals, and accelerated development [1]. Each region also faces the same challenges and is characterized by abundant natural resources on land and water. These natural resources are of interest and are included in the progress of a country, specifically the regions. Therefore, various forms of productive and multi-purpose use should be identified, planned, and developed. Varied use of forest resources is also important for conservation and resistance to pests and climate shocks [2]. This includes tourism activities as a more sustainable use of forest services. Thus, it is hoped that nature can be a solution for the community's economy and the conservation of nature itself.

As the main stakeholder, communities around forests are important parties to pay attention to in natural resource management. If the community obtains the benefits that can

be felt, then various forms of participation can be developed [3]. If these conditions are met, according to [4], local governments can encourage community participation in activities such as trade, business exhibitions, various cultural festivals and museums, organizing sports and art attractions, and investing in various types of businesses based on ecotourism. Collaborative ecotourism management is also a demand in sustainable ecotourism management. The involvement of stakeholders both locally and internationally will have a positive impact on their concern about how ecotourism destinations can be developed and managed sustainably [5]. Participation and involvement of local communities as well as the application of a responsible ecotourism model cannot be ignored, so that the goals of sustainable ecotourism can be realized [3].

Indonesia, specifically North Sumatra Province, has Tangkahan Ecotourism managed by local communities and has been around for a long time [6]. The area has become one of the leading tourism destinations in the province, locally, nationally, and internationally. This ecotourism was opened in 2001 and inaugurated in February 2004. The object is an ecotourism area with excellent local community participation in nature conservation. These nature tourism activists are precisely communities that previously relied heavily on the economy of the forest in harmful ways, such as illegal logging and hunting. The presence of ecotourism activities has become a solution for the community's economy that is in line with the principles of sustainability. Additionally, the region demonstrates how ecotourism growth may significantly protect the 17,000-hectare Gunung Leuser National Park (GLNP) area in North Sumatra [7].

The intangible benefits of forests cannot be assessed using a market system, and several users are unaware of these benefits. There is still a lack of appreciation for environmental benefits in the form of scenic beauty. Landscape beauty can be enjoyed and used by humans through nature tourism [8]. Efforts should be made to develop the form and management of its utilization to increase the value of the benefits. This study is necessary to ensure that the planning for the development of Tangkahan Ecotourism in Gunung Leuser National Park can be truly effective and provide significant benefits for the welfare of the community. This is because, according to [4], the benefits to the surrounding community will greatly affect the support of local communities for the development of sustainable tourism. These benefits should include both material and non-material domains.

People believe that the development of ecotourism can produce significant economic benefits for them [3]. The condition is that they must be able to play key roles both in the decision-making process and in the formulation of the direction of ecotourism management. For this reason, studies related to the economic benefits of Tangkahan ecotourism management are important to prove how much these economic benefits are manifested. Aspects that are also important to study are the characteristics and behavior of visitors. Therefore, this study analyzes the economic value of nature tourism objects, the intensity of visits, the tourist interest in revisiting, and the affecting factors in the Tangkahan Ecotourism Area, Langkat Regency, North Sumatra, Indonesia. According to [9], feedback from visitors is very helpful for ecotourism destination management institutions to determine the priorities and directions of wisdom in the development of tourist attractions. The importance of visitor returns is also related to competition and the application of the principles of sustainable ecotourism management.

## 2. Materials and Methods

The study was conducted in Tangkahan Ecotourism, Namo Sialang and Sei Serdang Villages, Batang Serangan Sub-district, Langkat Regency, North Sumatra Province, Indonesia (Figure 1).

Primary data were collected through questionnaires and field observations, while secondary data were collected from various institutions, especially Tangkahan Ecotourism managers. Quoted accidental sampling technique was used in collecting primary data [10].

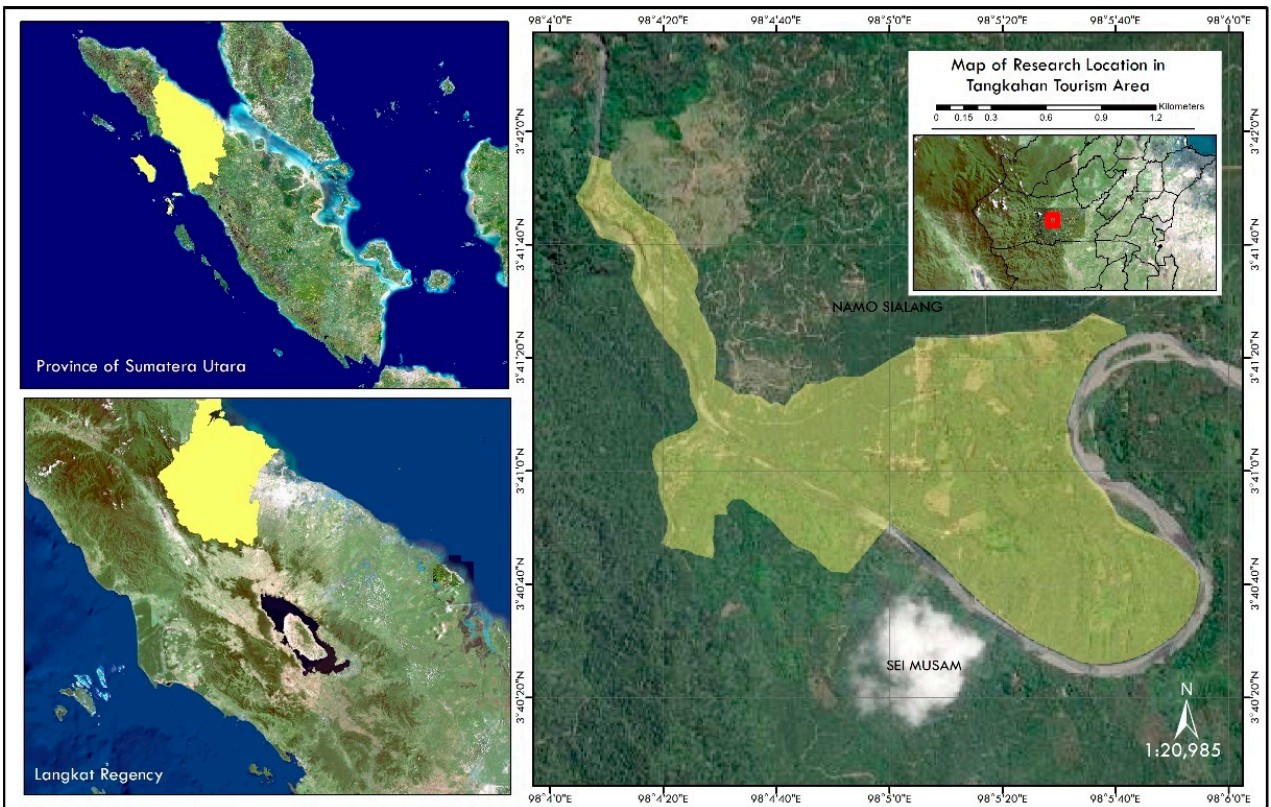

**Figure 1.** Tangkahan Ecotourism Location, Gunung Leuser National Park, North Sumatra, Indonesia (03°41′0″ N and 98°04′28.2″ E).

Secondary data collected are the characteristics of visitors and tourist attractions. The characteristics of visitors include the number of visitors each year, the number of inhabitants, the area of origin, as well as the number of inhabitants of the visitor's home zone. The characteristics of tourist attractions include location, area, biological conditions, tourism potential, accessibility, and recreational facilities.

The number of samples was determined by the Slovin formula, referring to the number of visitor populations of Tangkahan Ecotourism. The Slovin formula according to [11–13] is as follows:

$$n = \frac{N}{1 + N(e)^2}$$

Description:

$n$ = number of samples
$N$ = number of population
$e$ = error tolerance (0, 1)

Refers to the total population of 31,200 people/year according to Tangkahan Tourism Institute. The sample taken was 99.68 (increased to 100 people). The Slovin formula in determining the sample for nature tourism was also used by [14] in Natsepa Beach, Maluku Province; [15] in Carocok Painan Beach, West Sumatra Province; [16] in Ciwidey, West Java; and [17] in Gunung Ciremai National Park.

### 2.1. Travel Cost Analysis

A zoned travel cost approach can be used to estimate the economic value of ecotourism [18–20]. The use of the travel cost method for the valuation of tourism objects is also used by [21] in Bozcaada (Turkey), [22] in Valencia (Spain), and [23] in Taman Tasik Cempaka (Malaysia). The stages in calculating the economic value of ecotourism [24–28] are as follows:

2.1.1. Calculating the Number of Visitors from Each Origin Area (Zone) Based on Interviews with Respondents

$$Zi = Pi \times \sum Y$$

Description:

Zi: Number of visitors zone i
Pi: Percentage of zone visits i
$\sum Y$: Total visits

2.1.2. Determining the Average Travel Cost of the Total Travel Costs Incurred during Travel or Recreational Activities

$$BPR = TR + KA + TK + LL$$

Description:

BPR: Average travel cost (IDR/person)
TR: Transportation cost (IDR/person)
KA: Consumption and accommodation cost during the trip (IDR/person)
TK: Ticket cost (IDR/person)
LL: Other costs (IDR/person)

2.1.3. Determining the Average Travel Cost of Zone *i*

$$Xli = \frac{\sum BPi}{Ni}$$

Description:

Xli: Average travel cost of zone i
*Bpi*: Travel cost of the sample
*Ni*: Total population of zone i

2.1.4. Determining the Visit Rate per 1000 People in Zone i in One Year

$$LKi = \frac{\sum JPi}{\sum JPT} \times 1000$$

Description:

LKi: Visit rate of visitors in zone i
JPi: Number of visitors in zone i
JPT: Total population in zone i

2.1.5. Determining the Total Economic Value (NET), Obtained from the Following Formula

$$NET = \text{Average Travel Cost} \times \text{Average Number of Visitors}$$

2.2. *Analysis of Factors Affecting Economic Value, Intensity of Visits, and Tourist Interest in Revisiting*

To determine the socioeconomic factors that influence the intensity of travel visits, multiple linear analysis is used. The multiple linear regression models used are:

$$Y_1 = \alpha_0 + \alpha_1 X_1 + \alpha_2 X_2 + \alpha_3 X_3 + \alpha_4 X_4 + \alpha_5 X_5 + \alpha_6 X_6 + e$$

$$Y_2 = \beta_0 + \beta_1 X_1 + \beta_2 X_2 + \beta_3 X_3 + \beta_4 X_4 + \beta_5 X_5 + \beta_6 X_6 + \beta_7 X_7 + \beta_8 Y_1 + e$$

$$Y_3 = \gamma_0 + \gamma_1 Y_1 + e$$

Description:

$Y_1$ = Travel Cost Value (individual)

$Y_2$ = Intensity of visits (frequency of visits up to the time of the study)

$Y_3$ = Tourist Interest in Revisiting

$\alpha_i$, $\beta_i$, $\gamma_i$ = Regression coefficient of independent variable

$X_1$ = Visitor Age

$X_2$ = Education Level

$X_3$ = Income Level

$X_4$ = Distance from Object

$X_5$ = Number of Members

$X_6$ = Travel Time

$X_7$ = Information Acquisition

In order to produce unbiased data, multiple regression analysis models are evaluated by econometric evaluation with classical assumption tests. Multicollinearity, heteroscedasticity, and autocorrelation tests were performed and met all assumptions. A Likert scale approach is used to measure various ordinal variables with a range of 1–5 [29]. It is used to measure visitors' conditions, attitudes, and opinions. The most positive opinion and in line with the theoretical assumptions are given a score of 5 (maximum), and the most negative opinion is given a score of 1 (minimum). After the data are obtained from the Likert scale, the validity and reliability tests are conducted to determine the validity and consistency of the received data.

### 2.3. Overview of Study Location

Tangkahan is developed as an ecotourism area located on Gunung Leuser National Park (GLNP) border. The area of the Tangkahan Ecotourism is ±103 hectares, which is divided into village and forest with an area of 18,526 ha and 17,653 ha, respectively [30]. Tangkahan is at an altitude of 130–200 m above sea level. The area's topography consists of hilly areas with varying slopes (45–90°). The Tangkahan area is located at the confluence of the Buluh and Batang Serangan rivers. This area has unique natural formations, beautiful landscapes, hot springs, waterfalls, caves, cliffs, high diversity of flora and fauna, and tropical rain.

### 2.4. Socioeconomic Characteristics of Visitors

An overview of the profile of visitors who travel to Tangkahan Ecotourism is obtained from the characteristics of respondents. The majority of visitors are domestic tourists from the area with a distance of 1–4 h, such as Stabat, Binjai, Medan, and several areas in North Sumatra. In certain seasons, it is also visited by many foreign tourists. The figures below show the distribution of respondents based on the type of tourists and their origin area (Figure 2). For the category of origin area, visitors consist of 20% and 80% of domestic and foreign tourists.

The characteristics of the visitors observed include age, gender, education level, occupation type, and income level. Visitors from the area are dominated by women (42% males and 58% females). A similar result was reported by [31], where 57% of Tangkahan visitors were female. This is because women prefer to spend recreational time with their friends. More women engage in tourism activities for various purposes [32–34]. The domination of women also occurs in families within the area. The average level of education of visitors is quite good. Most visitors have at least 12 years of education. Most visitors have received an education for at least 12 years. The majority of visitors' education levels are senior high school level (57%), 41% of visitors are undergraduate educated, and 2% have a master's degree. The occupation type of respondents is very diverse, with the largest proportion being students (33%), followed by entrepreneurs (17%) and private employees (15%).

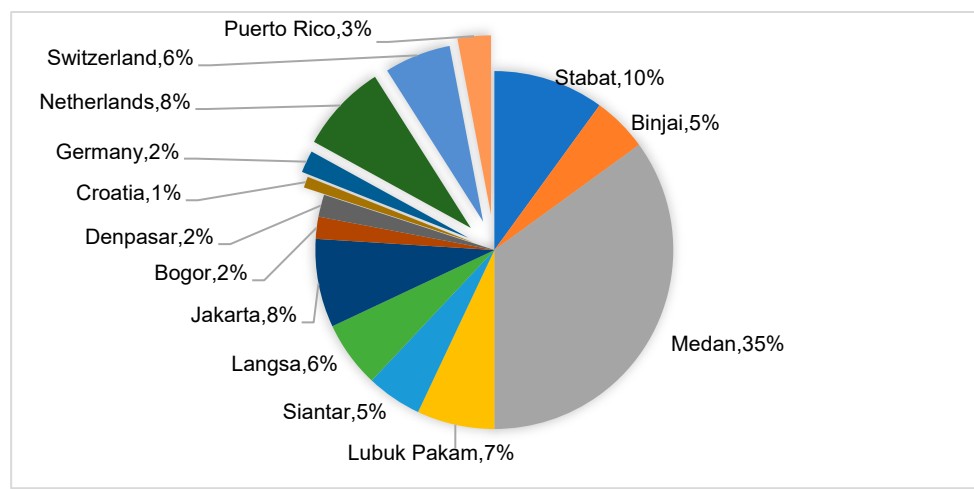

**Figure 2.** Distribution of Tangkahan Ecotourism respondents based on the origin of visitors.

The income level of visitors varies from the lowest to the highest segment (Figure 3). Theoretically, the level of visitor income will affect expenses during tourist visits. The allocation of expenditures includes transportation, consumption, accommodation, and other costs. Income is also expected to influence the choice of tourism objects to be visited. These data are consistent with the occupation type data, where most visitors are students. Students do not have income at that age, and are still supported by their parents, including budget allocations for tourism purposes.

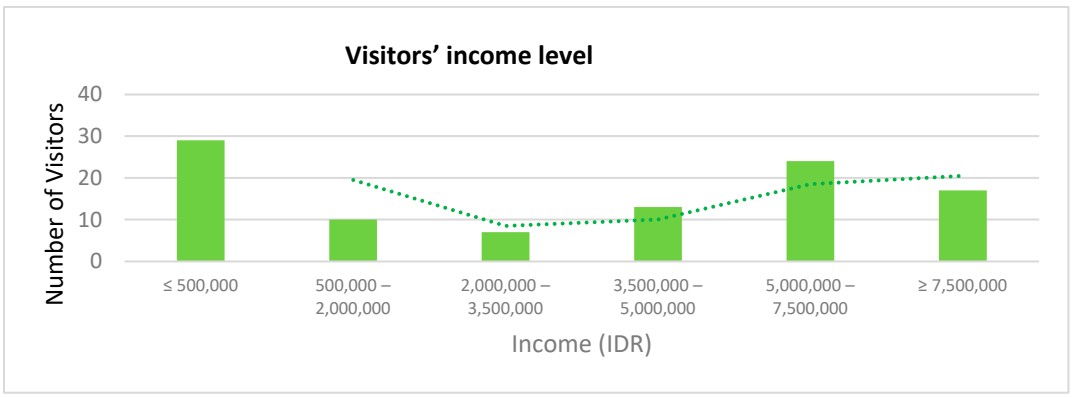

**Figure 3.** Graphic of visitors' income level.

## 3. Results and Discussion

### 3.1. Valuation of Economic Value of Tangkahan Ecotourism

Calculating the value of the intangible benefits of a recreation area can be done through the approach of the travel cost method. Furthermore, the total value included in the travel cost is the round trip cost plus the monetary value of the time spent on travel and recreational activities [35]. This method is widely used in various nature-based tourism objects such as Lake Limboto [36], Kalibiru [37], Parangtritis [38], Muara Angke [39], Batu Karas Beach, Pangandaran [40], Punti Kayu, Palembang [41,42], Thousand Islands [43], Ujung Genteng, Sukabumi [44], and others. The economic assessment of the Tangkahan Ecotourism area collected includes information about the origin of visitors, the cost of round-trip visitor travel, consumption costs during tourist visits, ticket fees for entrance to tourist objects, and other costs that must be paid (documentation, parking, storage, guides, supporting equipment, and documentation fees).

Table 1 shows the highest average travel cost value comes from Zagreb at IDR 14,600,000/visit, while the lowest comes from Stabat as the origin area of visitors closest to

this tourism object at IDR 52,700/visit. The average travel costs that should be incurred from all visitors and all origin areas/countries of visitors is IDR 4,181,786/visit/person.

**Table 1.** Recapitulation of visitor data based on average travel cost.

| No | Visitor Origin | $\bar{x}$ Transportation Cost (IDR) | $\bar{x}$ Consumption Cost (IDR) | Ticket Cost (IDR) | $\bar{x}$ Other Cost (IDR) | $\bar{x}$ Travel Cost (IDR) |
|----|---------------|-----------------|-----------------|-----------|------------|------------|
| 1 | Stabat | 21,000 | 22,900 | 3000 | 5800 | 52,700 |
| 2 | Binjai | 38,000 | 28,000 | 3000 | 9000 | 78,000 |
| 3 | Medan | 56,971 | 31,343 | 5343 | 8257 | 101,914 |
| 4 | Lubuk Pakam | 42,857 | 58,857 | 3000 | 10,000 | 114,714 |
| 5 | Siantar | 51,000 | 49,000 | 3000 | 8800 | 111,800 |
| 6 | Langsa | 83,333 | 20,833 | 3000 | 7500 | 114,667 |
| 8 | Jakarta | 2,193,750 | 48,750 | 26,875 | 174,625 | 2,444,000 |
| 9 | Bogor | 3,000,000 | 200,000 | 100,000 | 250,000 | 3,550,000 |
| 10 | Denpasar | 3,494,000 | 80,000 | 100,000 | 107,500 | 3,781,500 |
| 11 | Croatia | 12,400,000 | 450,000 | 250,000 | 1,500,000 | 14,600,000 |
| 12 | Germany | 9,300,000 | 175,000 | 250,000 | 1,250,000 | 10,975,000 |
| 13 | Netherlands | 7,756,250 | 302,500 | 250,000 | 1,243,750 | 9,552,500 |
| 14 | Switzerland | 8,086,666 | 326,666 | 250,000 | 526,666 | 9,190,000 |
| 15 | Puerto Rico | 7,460,000 | 200,000 | 250,000 | 150,000 | 8,060,000 |
| Average after weighting each origin cluster of visitors | | | | | | 4,181,786 |

The costs incurred by respondents, according to the results of data recapitulation, in carrying out tourist activities (based on total travel costs), obtaining the economic value of the existence of Tangkahan Ecotourism, were IDR 72,708,168,000, -/year (equivalent to US $505,514.6). This value is obtained from the multiplication between the average value of visitor travel costs (IDR 4,181,786/visit) and the average number of tourist visits in 1 year. The average number of annual visits (31,200 people/year) was taken from data for the last three years from 2016 to 2018, based on data retrieved from the Tangkahan Tourism Institute in 2019.

The average travel cost of IDR 4,181,786 per visit is already high because Tangkahan is an ecotourism special interest tourism. Therefore, it selects visitors and attracts special groups of interested people willing to pay more to enjoy the specificity of a nature tourism area. In Bozkaada, Turkey, the economic value is TL 21,795,492.32 [21], and the travel cost value per person is TL 4.80 per travel or TL 110 per season [15]. Furthermore, the economic value of tourism in Kaziranga National Park is INR 773.45 million (INR 187.6 per visitor) [45]. Malaysia has a net economic value of MYR 6.2/visit/person [23].

This economic value is far greater than the revenue obtained from receiving an entrance ticket of IDR 3000/visit. The total revenue obtained by the manager from ticket sales is only IDR 71,814,000/year. This illustrates that the economic value of natural resources is from direct revenue and all benefits received by all parties related to the management/utilization of these natural resources. Therefore, this study uses a total travel cost approach to calculate the entire value received by the parties involved in Tangkahan Ecotourism.

In addition to material benefits, the existence of ecotourism also contributes to the welfare of communities around the forest in a non-material form. The important determinants of the quality of human life include the material and non-material domains [4]. The economic benefits of the existence of Tangkahan Ecotourism have an impact on material aspects, while changes in socio-cultural aspects such as increasing respect for the environment, strengthening community institutions in ecosystem management, changes in lifestyle and livelihoods, lack of potential disaster threats, and a sense of security are positive non-material impacts. These non-material benefits are inseparable from the development of Tangkahan Ecotourism as a form of active involvement of communities around national parks in managing forest ecosystems. Related to the above, as a result, [46] reported that the participating community in the development of Tangkahan Ecotourism has made a major contribution to the conservation of the Gunung Leuser National Park area. For more

than 20 years the Tangkahan Ecotourism area has been running under the auspices of an institution, namely the Tangkahan Tourism Institute (LPT) [11].

### 3.2. Factors Affecting Economic Value

An econometric evaluation with classical assumption tests, including multicollinearity, heteroscedasticity, and autocorrelation tests, was conducted before the regression analysis. The multicollinearity test results show that the VIF value is less than 10, and the tolerance is less than 1 for all study variables. Based on the results of the multicollinearity test, the value of Variance Inflation Factor (VIF) was less than 10, and the value of Tolerance was less than 1 for all variables studied, so it was concluded that there was no multicollinearity in the regression. The heteroscedasticity test using graphical aids also shows an even distribution of points above and below the value 0. The autocorrelation test using the Durban Watson Test showed a DW value close to a value of 2. In general, the test results stated that there was no violation of assumptions so it was feasible to proceed to the next stage of testing [47–49].

Based on Tables 2 and 3, the regression equation obtained is $Y_1 = 8.295 + 0.192X_1 - 1.924X_2 + 1.518X_3 + 1.980X_4 - 0.733X_5 + 2.203X_6$. From the evaluation results of the test model, age, education, income, distance, number of members, and travel time significantly affect individual visitors' travel costs with a coefficient of 57.1%. The factors that partially have a significant effect on the economic value of the Tangkahan Ecotourism area are education, income, distance of objects from the origin of visitors, and the travel time of visitors to reach tourist sites.

**Table 2.** Simultaneous test results (F-test) for travel expenses value.

| | Model | Sum of Squares | Df | Mean Square | F | Sig. |
|---|---|---|---|---|---|---|
| | Regression | 324.812 | 6 | 54.135 | 38.260 | 0.000 [b] |
| 1 | Residual | 131.589 | 93 | 1.415 | | |
| | Total | 456.402 | 99 | | | |

Dependent variable: the value of travel expenses; [b] Predictors: (constant), visitor age, education level, income level, distance from area, number of members, length of travel, and acquisition of information.

**Table 3.** Partial test results (*t*-test) for travel expenses value.

| Variable | B | Std. Error | Beta | T | Sig. |
|---|---|---|---|---|---|
| (Constant) | 8.295 | 1.011 | | 8.208 | 0.000 |
| Age | 0.192 | 0.223 | 0.051 | 0.860 | 0.392 |
| Education | −1.924 | 0.875 | −0.160 | −2.199 | 0.030 |
| Income | 1.518 | 0.243 | 0.491 | 6.240 | 0.000 |
| Distance | 1.980 | 0.743 | 0.275 | 2.666 | 0.009 |
| Number of Members | −0.733 | 0.562 | −0.077 | −1.305 | 0.195 |
| Travel Time | 2.203 | 0.760 | 0.309 | 2.899 | 0.005 |
| R Adjusted | 0.844 | | | | |
| $R^2$ | 0.712 | | | | |

Education has a positive effect with a negative coefficient on the travel cost. Therefore, visitors with higher levels of education spend less on travel costs. This is not consistent with the theoretical assumption that groups with higher education should be willing to spend more [50,51]. Based on field observations, the lower travel cost of the more educated group is due to their better ability to organize visits and reduce travel costs individually. These efforts include using more mass transportation facilities, more planned management of visit activities, and better access to information technology. Therefore, transactions can be conducted more efficiently through online-based ordering and transaction services.

Income has a significant and positive effect on travel costs. This is consistent with the theoretical assumption that higher-income visitors will spend more to enjoy nature

tourism areas, as reported by [52] in Kodagu District, India, [53] concerning Nature-Based Tourism (NBT), and [54] in Kalam Valley of Khyber Pakhtunkhwa, Pakistan. Distance and travel time are synchronous factors, where the distance is directly proportional to the travel time from the visitor's origin to the location of a tourism site. These two variables have a significant and positive effect; therefore, the distance and the travel time are directly proportional. Visitors need to spend more to enjoy Tangkahan nature tourism areas [55].

### 3.3. Factors Affecting Intensity of Visits

According to descriptive statistical data, tourists visited Tangkahan 1.6 times on average. This illustrates that many visitors repeat visits to Tangkahan tourism areas due to the good impression obtained from various aspects. The intensity of visits to Tangkahan is low/moderate/high compared to other tourism areas studied.

The intensity of tourist visits in Tangkahan illustrates that there are still demands to improve the impression of visitors by increasing aspects that significantly affect the intensity of tourists. Improving service quality through important elements is expected to be more effective [56–59].

Table 4 explains that the F-count value is greater than the F-table. Therefore, the independent variable simultaneously has a significant effect on the dependent variable (intensity of tourist visits). This shows that travel cost, age, education level, income, distance from residence to tourism objects, number of members in the group, travel time to be taken, and the acquisition of information related to tourism sites affect the intensity of visiting Tangkahan nature locations. The high coefficient of determination in this regression model (93.2%) indicates that the eight variables above can simultaneously explain almost all changes/variations in the intensity of visits.

**Table 4.** Simultaneous test results (F-test) for visit intensity.

| | Model | Sum of Squares | Df | Mean Square | F | Sig. |
|---|---|---|---|---|---|---|
| | Regression | 14.489 | 8 | 1.811 | 17.440 | 0.000 [b] |
| 1 | Residual | 9.450 | 91 | 0.104 | | |
| | Total | 23.940 | 99 | | | |

Dependent variable: intensity of visits; [b] Predictors: (constant), information, number of members, education, age, travel time, income, distance, travel cost.

The F-test has not shown which independent variables directly and significantly affect the dependent variable (intensity of visits). Therefore, it is continued with multiple linear regression analysis to determine which variables significantly affect the dependent. Based on Table 5, the regression equation obtained is $Y_2 = 1.570 + 0.072X_1 - 0.140X_2 + 0.415X_3 - 0.185X_4 - 1.457X_5 - 0.244X_6 - 0.026X_7 - 0.066X_8$.

**Table 5.** Partial test results (*t*-test) for visit intensity.

| Variable | B | Std. Error | Beta | T | Sig. |
|---|---|---|---|---|---|
| (Constant) | 1.570 | 0.361 | | 4.352 | 0.000 |
| Travel Cost | 0.072 | 0.029 | 0.315 | 2.475 | 0.015 |
| Age | −0.140 | 0.064 | −0.162 | −2.194 | 0.031 |
| Education | 0.415 | 0.244 | 0.151 | 1.705 | 0.092 |
| Income | −0.185 | 0.079 | −0.262 | −2.357 | 0.021 |
| Distance | −1.457 | 0.209 | −0.882 | −6.953 | 0.000 |
| Number of Members | −0.244 | 0.154 | −0.111 | −1.587 | 0.116 |
| Travel Time | −0.026 | 0.216 | −0.016 | −0.120 | 0.904 |
| Information | −0.006 | 0.069 | −0.006 | −0.087 | 0.931 |
| R Adjusted | 0.778 | | | | |
| $R^2$ | 0.605 | | | | |

Simultaneously, these predictor variables have a significant effect on the intensity variables of visits. These independent variables significantly affect the dependent (intensity of visits). However, the partial test shows that not all independent variables affect the intensity of visits. The test results showed that of the seven socioeconomic variables observed in this study, there were only two variables that had a significant effect (using $\alpha = 0.05$) on the intensity of tourist visits to the Tangkahan Ecowista area. The two variables are the distance and the number of members with a negative sign. It shows that visits will increase as the distance decreases and the number of members decreases. Distance is very influential in the selection of tourism objects. Another study stated that 4 out of 10 independent variables tested directly affected tourist revisits, such as safety and security, description of destinations, infrastructure, and price [60].

This study found that the farther the tourism site from the residence, the less intense the tourist visits. Visitors prefer tourist destinations that are closer to their homes [40], indicating distance significantly affects tourist visits, specifically costs and benefits [61–63]. People closer to tourism areas are more supportive of tourism activities than those far away. The distance factor is often a barrier to tourist visits to nature tourism areas; therefore, it is necessary to support adequate regional transportation infrastructure to minimize the distance factor with good access quality to shorten travel time.

The greater the number of members in the visiting group, the less the intensity of visits to Tangkahan. This shows the tendency of tourists to enjoy visits with fewer members. The major attractions of nature tourism are rivers and landscapes; thus, people prefer to experience them in smaller groups. Therefore, it is necessary to have facilities and tourism object designs that support small and personal group-based activities to enjoy their privacy more.

The other five variables, such as travel cost, age, education, income, travel time, and information acquisition, have no significant effect (using $\alpha = 0.05$) on the intensity of tourist visits. In general, traveling can be conducted on weekends and national holidays. For foreign tourists, visits are made during seasonal holidays. During that period most people will plan trips to tourist attractions that present recreational attractions. The factors of per-street cost, age, education, income, length of travel, and acquisition of information tend not to be significant considerations for tourists, so that it has a negligible influence on the intensity of tourist visits.

*3.4. Tourist Interest in Revisiting*

The average score of interest in revisiting the Tangkahan Ecotourism area is 4.21 (on a scale of 5). This total score is in the very high category (slightly past the very high-class limit); therefore, most tourists express interest in revisiting. This answer does not depend on the intensity of visits to describe the visitors' impression of the area. The interest in revisiting is increases when it is close to a score of 5 (total score of 500), where all visitors are willing to come back for a tour according to Pareto terms. The interest score in revisiting by 4.21 is very high compared to other tourism attractions. According to [64], there is a total of 2948 people willing to revisit the Ciwangun Indah Camp. In Banyuwangi, the Effectiveness of Tourism Destination Advertisements on Interest in Revisiting had a score of 3.66 [65].

This study shows that satisfaction is a factor that has a direct significant influence on short-term return visit intentions, while the novelty of tourist attractions is a factor that has a significant effect on medium- to long-term return visit intentions [66]. In the United Arab Emirates, satisfaction affects the interest in revisiting [67]. The impressions of visitors which are positive but have not reached the highest score illustrate that even though Tangkahan nature tourism areas are quite attractive to visitors, some aspects of service need to be addressed. Improvements carried out effectively are expected to increase the average score of the interest in revisiting to close to 5. The aspects developed should be prioritized to affect the interest/willingness/intention to revisit significantly.

A simple model of estimating the tourist interest in revisiting ($Y_3$) is obtained by using travel cost ($Y_1$) as the independent variable (Table 6). The regression analysis results show

that travel cost significantly affects the tourist interest in revisiting (Table 7). However, this regression model is simple, with a sufficient coefficient of determination at 43%. This also tends to be different from the theoretical assumption that the cost is a factor inhibiting tourist interest in revisiting. Empirically, this is possible because Tangkahan natural tourism is a special interest tourism, so the cost factor tends to be in elastic. The average travel cost to enjoy this tourist area is relatively low; hence, it has not been considered a factor that becomes a negative factor for visiting. However, it is still necessary to conduct a special study to explain this matter further.

**Table 6.** F-test results for tourist interest in revisiting.

| | Model | Sum of Squares | Df | Mean Square | F | Sig. |
|---|---|---|---|---|---|---|
| | Regression | 0.156 | 1 | 0.156 | 5.400 | 0.022 [b] |
| 1 | Residual | 2.835 | 98 | 0.029 | | |
| | Total | 2.991 | 99 | | | |

Dependent variable: interest in revisiting; [b] Predictors: (constant), travel cost.

**Table 7.** Linear regression results for tourist interest in revisiting.

| Variable | B | Std. Error | Beta | T | Sig. |
|---|---|---|---|---|---|
| (Constant) | 2.288 | 0.102 | | 22.406 | 0.000 |
| Travel Cost | 0.019 | 0.008 | 0.229 | 2.324 | 0.022 |
| R Adjusted | 0.229 | | | | |
| $R^2$ | 0.052 | | | | |

## 4. Conclusions

The economic value of the Tangkahan Ecotourism area with the zoned travel cost method is IDR 72,708,168,000/year. On average, tourists have visited Tangkahan 1.6 times. Simultaneously, the factors of age, education, income level, distance, number of members, and travel time significantly affect the value of the individual travel costs of visitors, with a regression model $Y_1 = 8.295 + 0.192X_1 - 1.924X_2 + 1.518X_3 + 1.980X_4 - 0.733X_5 + 2.203X_6$. Education, income, distance, and travel time partially affect the economic value. This is a reference for the right promotional segmentation policy in order to increase the economic value of the existence of Tangkahan Ecotourism. The tourist interest in revisiting Tangkahan nature tourism objects is very high (score 4.21). Generally, travel cost, age, education level, income level, distance, number of members, travel time, and information acquisition significantly affect the intensity of visits with regression model $Y_2 = 5.975 + 1.040 \times 10^{-8}X_1 - 0.097X_2 + 0.267X_3 + 0.121X_4 - 0.723X_5 - 0.515X_6 - 0.116X_7 - 0.190X_8$. Distance, number of members, and travel cost significantly affect the intensity of tourist visits to the Tangkahan Ecotourism area. Ecotourism managers must improve accessibility infrastructure, increase comfort for visitors with large groups, and minimize travel costs to increase tourist interest in visiting again. Socio-cultural variables and visitors' assessment of ecotourism sustainability aspect are recommended to be involved in the next research.

**Author Contributions:** Drafting concepts, A.P. and D.R.N.; data analysis, A.P., D.R.N. and H.L.G.; format adjustment, H.L.G.; fund-raising, A.P.; clarification of data, A.P., M.E. and H.L.G.; method design, A.P.; field equipment, H.L.G.; software, S.R.; supervision, S.R. and M.E.; visual display, A.P. and H.L.G.; writing and editing drafts, A.P. and H.L.G. All authors have read and agreed to the published version of the manuscript.

**Funding:** This study received no external funding.

**Institutional Review Board Statement:** The study was conducted in accordance with the Declaration of Helsinki, and approved by the Ethics Committee of Universitas Sumatera Utara (No. 396/KEPK/USU/2022, date of approval 22 April 2022).

**Informed Consent Statement:** Informed consent was obtained from all subjects involved in the study.

**Data Availability Statement:** Not applicable.

**Acknowledgments:** The authors would like to thank the Universitas Sumatera Utara for funding this research. We also extend our gratitude to The Manager of Gunung Leuser National Park (TNGL) and the Tangkahan Tourism Institute (LPT) for access, data, and explanations in the field. We also express our gratitude to visitors to Tangkahan Ecotourism for their willingness to be respondents.

**Conflicts of Interest:** The authors declare no conflict of interest.

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
