# Peer review of "Examining the Economic Value of Tourism and Visitor Preferences: A Portrait of Sustainability Ecotourism in the Tangkahan Protection Area, Gunung Leuser National Park, North Sumatra, Indonesia"

_sustainability, doi:10.3390/su14148272_

Round 1
Reviewer 1 Report
This study try to analyze the economic value of tourism and the preferences of tourists to revisit, along with the factors that influence them, where these conditions can be an illustration of the sustainability of Tangkahan ecotourism. The travel cost method is used to determine the economic value of Tangkahan Ecotourism environmental services. The multiple linear regression is used to analyze the factors that affect the economic value, intensity of visits, and interest in revisiting. The results showed that Tangkahan ecotourism has a relatively high economic value, supported by the intensity and interest of tourist visits. The topic of this manuscript is fit very well of the Journal’s name. Hopefully, this manuscript can attract a wider audience. The reviewer recommend to accept this manuscript if the author correct the following topics :
· The research uniqueness and contribution should be addressed in the introduction part of this manuscript.
· The author should mention about Figure 1 in the text.
· The Reference should be revised throughout. The software like Endnote should be used to compose the reference.
· Ref. no. 16 : “Lamtrakul, P.; K. Teknomo., K. Hokao. Public Park Valuation Using Travel Cost Method. Proceedings of the Eastern Asia Society for Transportation Studies, 1249 – 1264 (2005).” Should be “Iamtrakul, P.; Teknomo K.; and Hokao K………
Author Response
Thank you for your review.
Thank you for your review. We have made an earnest effort to understand your suggestions for improvement, and we have tried to improve them. Your suggestions and input mean a lot to the progress of my article. we hope that our current article will get better. I've also corrected the corrections from academic editors.
For more revisions, please see the attachment.
Best regards,
Agus Purwoko

Reviewer 2 Report
Dear authors,
Your paper is well written and is interesting to read. In my view, it would benefit from a few improvements that I'll try to point out in the following lines:
1. Because your paper has no Literature Review section, the Introduction, which has less than one page in length, should be extended in order to include the relevant literature on tourism in natural areas and the need for tourism planning and for the involvement of the stakeholders, especially the local communities in order to increase the guest experience and satisfaction and to assure local sustainability. Please consider improving the Introduction. Maybe the following (and other) papers on the topic could be used to that end:
https://doi.org/10.3390/su132313302
https://doi.org/10.1080/10548408.2019.1689224
https://doi.org/10.18089/tms.2021.170201
https://doi.org/10.1016/j.jhtm.2020.09.005
2. In the Results section, some descriptive text could be deleted as it is irrelevant and boring for an international reader (ex. Geographically, it is located at 030 41'01" North Latitude and 980 04' 28.2" East Longitude [26]. Administratively, this area is included in Namo Sialang and Sei Serdang Villages, Batang Serangan Sub-district, Langkat Regency, North Sumatra Prov-ince, Indonesia. Tangkahan is at an altitude of 130-200 meters above sea level. The area's topography consists of hilly areas with varying slopes (45-900).
The Tangkahan area is located at the confluence of river Buluh and Batang Serangan. This area has unique natural formations, beautiful landscapes, hot springs, waterfalls, caves, cliffs, high diversity of flora and fauna, and tropical rain. Based on data from the Tangkahan Tourism Institute, this area has the following boundaries: Northside : Oil palm plantation owned by PTPN II Kuala Sawit; Southside : Oil palm plantation owned by PT. Permana Doubles; Eastside : Kuala Buluh Hamlet; West side : Gunung Leuser National Park (TNGL)»
3. The area of origin of the tourists is given as their home cities/town and not countries: ex. Figure 2: Geneva 2%; Bern 4%; Haarlem 3%; Vlissingen 2%; Leiden 3%, etc. Why not Switzerland 6%, The Netherlands 8%, etc? At least for the international tourists.
4. Conclusions give a clear and objective answer to the objectives. Very well done. However, I miss the theoretical implications (the contributions of your study for the advancement of the topic) and practical Implications (i.e. how your study can improve tourism and sustainability in similar regions). You could also add the study's limitations and recommendations for future research.
Wish you much success!
Author Response

(The authors gave the same response as above.)

Round 2
Reviewer 2 Report
Dear authors,
The new version has significantly improved. Thanks
This manuscript is a resubmission of an earlier submission. The following is a list of the peer review reports and author responses from that submission.